# Biorefinery of Lignocellulosic and Marine Resources for Obtaining Active PVA/Chitosan/Phenol Films for Application in Intelligent Food Packaging

**DOI:** 10.3390/polym17010082

**Published:** 2024-12-31

**Authors:** Mary Isabel Lopretti Correa, Diego Batista-Menezes, Stephany Cunha de Rezende, Arantzazu Santamaria-Echart, Maria-Filomena Barreiro, Jose Roberto Vega-Baudrit

**Affiliations:** 1Laboratory of Nuclear Techniques Applied to Biochemistry and Biotechnology, Nuclear Research Center, Faculty of Sciences, Universidad de la República, Mataojo 2055, Montevideo 11400, Uruguay; mlopretti@gmail.com; 2National Nanotechnology Laboratory, National Center for High Technology, Pavas, San José 10109, Costa Rica; dbatista@cenat.ac.cr; 3Centro de Investigação de Montanha (CIMO), Instituto Politécnico de Bragança, Campus de Santa Apolónia, 5300-253 Bragança, Portugal; rezendes@ipb.pt (S.C.d.R.); asantamaria@ipb.pt (A.S.-E.); barreiro@ipb.pt (M.-F.B.); 4Laboratório Associado para a Sustentabilidade e Tecnologia em Regiões de Montanha (SusTEC), Instituto Politécnico de Bragança, Campus de Santa Apolónia, 5300-253 Bragança, Portugal; 5Escuela de Química, Universidad Nacional, Heredia 40101, Costa Rica

**Keywords:** lignocellulosic, chitosan, films, innovative packaging

## Abstract

This study focuses on the extraction of phenolic compounds from the fermentation of *Phanerochaete chrysosporium* and *Gloeophyllum trabeum*. The main goal was to synthesize phenol/chitosan microspheres and PVA films and characterized using FTIR, TGA, DSC, SEM, and mechanical tests to evaluate their physical, chemical, and mechanical properties for antimicrobial packaging applications. Homogeneous chitosan microspheres loaded with lignin-derived phenols were obtained, showing controlled release of antimicrobial compounds. The incorporation of phenolic microspheres into PVA/chitosan films resulted in significant improvements in mechanical properties: the films exhibited an elastic modulus of 36.14 ± 3.73 MPa, tensile strength of 12.01 ± 1.14 MPa, and elongation at break of 65.19 ± 5.96%. Thermal tests revealed that chitosan-containing films had enhanced thermal stability, with decomposition temperatures (T10) reaching 116.77 °C, compared to 89.28 °C for pure PVA. In terms of antimicrobial activity, PVA/chitosan/phenol films effectively reduced *Lactobacillus* growth and milk acidity, maintaining quality for up to 96 h at room temperature, outperforming controls with acetic acid and H_2_O_2_. The films also inhibit yeast growth for one week. In conclusion, phenols can be effective antimicrobial agents in dairy, but their use should be monitored. Additionally, PVA/chitosan-phenol films offer biodegradability, antimicrobial properties, and sustainability for diverse applications.

## 1. Introduction

Biorefinery has been proposed as a model for sustainable processes that use available biomass resources and renewable raw materials, aiming to minimize environmental impacts and produce various biological products [1,2]. In this context, a multi-product biorefinery should not have significant trade-offs or competition between food and non-food resources [3]. In natural ecosystems, fungi decompose and recycle various organic materials, such as lignocellulose, found in wood and agricultural waste [4,5]. Based on this concept, it is possible to use natural resources such as lignocellulosic biomass to extract and produce value-added compounds, such as edible and medicinal fungi, nanocellulose, chemical precursors, and biofuels. With the evolution of the biorefinery concept, exciting research has also been conducted on the diversification of biorefineries, utilizing different types of biomass, including marine-origin biomasses [6,7,8].

Regarding food packaging, the increasing demand for prepared foods, due to their taste, convenience, and nutritional value, has continuously renewed the importance of packaging to maintain food safety and extend the shelf life of prepared foods [9,10]. By applying an intelligent food packaging system, it is possible to quickly determine if a perishable product is safe to eat from its packaging, which enhances consumer food safety. The demand for active and intelligent packaging technologies is increasing globally, and warning labels are one of the preferred methods for implementing intelligent packaging [11,12,13]. However, incorporating these warning labels, such as freshness indicators or time-temperature indicators, into food packaging made from synthetic materials is challenging without affecting the quality and safety of the food or the environment. Therefore, developing bio-based food packaging as a platform technology to ensure food quality and safety is of practical importance, especially for packaging perishable products.

Chitosan, a biopolymer derived from chitin, has garnered significant attention due to its biodegradability, biocompatibility, and non-toxicity. Chitosan films are created by dissolving chitosan in acidic solutions, followed by casting and drying processes. These films exhibit excellent film-forming abilities, good mechanical properties, and antimicrobial activities, making them suitable for various applications in food packaging, biomedical fields, and wastewater treatment [14,15,16,17]. Polyvinyl alcohol (PVA) is a synthetic polymer known for its excellent film-forming properties, water solubility, and biocompatibility. PVA films are typically produced by solution casting, where PVA is dissolved in water, poured onto a substrate, and dried [18,19]. These films are used in applications ranging from packaging materials to medical and pharmaceutical industries, particularly in drug delivery systems and wound dressings [19,20]. On the other hand, phenolic groups interact with the polymer matrix, enhancing its stability and functionality. Phenol-modified chitosan/PVA films have shown great potential in applications requiring improved antimicrobial activity and structural integrity [21,22,23].

The combination of chitosan, PVA, and phenol results in films with synergistic properties. These composite films leverage the biocompatibility and antimicrobial properties of chitosan, the film-forming ability and flexibility of PVA, and the reinforcing and antimicrobial effects of phenol. The potential use of these films in advanced packaging, biomedical devices, and environmental applications is currently being studied.

Microspheres allow for the controlled release of phenolic compounds, ensuring a sustained antimicrobial effect over time. This is particularly useful in applications such as wound dressings, drug delivery systems, and surface coatings where prolonged protection is needed. Non-toxic and safe, the microspheres are suitable for applications in food preservation, cosmetics, and pharmaceuticals without the risk of toxicity. Phenolic compounds can be sensitive to environmental conditions such as light, temperature, and pH. Encapsulation in chitosan microspheres helps protect these compounds, improving their stability and effectiveness. In summary, chitosan microspheres with phenols offer a potent, safe, and versatile solution for antimicrobial applications across diverse fields, providing controlled release, biocompatibility, and enhanced germicidal and germistatic effects.

Fo the present work, PVA/Quito Sano/phenolic compounds membranes were chosen, after several previous works by the group where the obtaining of phenolic compounds with germicidal activity [24] was studied, several formulations of the membranes were studied, including the incorporation of micro capsules of phenolic lignin compounds [25,26], and the germicidal control activity was studied in specific cultures of bacteria, fungi, and yeasts, including some pathogenic bacteria such as Salmonella that can colonize food. These being chosen for their physical, chemical, mechanical, and in vitro microbiological control properties, they were tested with foods, studying their behavior and viability.

The main objective is to develop biodegradable films based on PVA, through the incorporation of chitosan and phenols, which are extracted from marine and lignocellulosic waste, respectively. Based on this, the aim is to use these compounds as value-added materials with intelligent properties to create the third generation of smart food packaging. These packaging materials are expected to exhibit natural antimicrobial activity and additional features such as low cost, light weight, and improved mechanical properties.

## 2. Materials and Methods

### 2.1. Strains Obtainment

*Phanerochaete chrysosporium* and *Gloeophyllum trabeum* strains were obtained from the TNA Laboratory in Biochemistry and Biotechnology of CIN, Faculty of Sciences of the University of the Republic, Uruguay. They were kept in potato dextrose agar (PDA) medium and incubated at 22 ± 2 °C for 5–8 days. For liquid fermentation, GPEL broth (glucose (20 g/L), yeast extract (2.5 g/L), and peptone (2.5 g/L)) adjusted to a pH of 4.5 with 0.1 N HCL and 0.1 N NaOH were used.

### 2.2. Phenolic Lignin Units

The FF units were produced in a semi-solid fermentation system for 30 days in a controlled chamber [27]. A 50 mL rice husk (25 g) was inoculated with 2 g of *Phanerochaete chrysosporium* and *Gloeophyllum trabeum* pellets. The semi-solid fermentation process was carried out in Rainbow solid fermentation units with humidity 60–70%; temperature 30 °C; pH 5.0–6.0. The activity of the inoculated fungi was monitored by measuring the enzymatic activity in the semisolid fermentation material. To do this, 2 g of inoculated rice husk was taken and suspended in 10 mL water. After 1 h, the enzymatic activity in the liquid was determined: lignin peroxidase at an average of 40 EU/mL and methoxyl hydrolase at an average of 10 EU/mL. Thirty days with the enzymatic action, a volume equal to that occupied by the solid material, 50 mL, was leached with physiological saline for 2 h and centrifuged. The extract was lyophilized.

### 2.3. Phenolic Compound Determintion

The lyophilized extract was redissolved in ethanol/water (80:20, *v*/*v*) to determine the phenolic compounds profiles by the Folling technique and was informed by mg/g of extract.

### 2.4. Synthesis of Phenol/Chitosan Spheres

For the preparation of the spheres, a 1.0% (*w*/*v*) chitosan (Sigma, Medium Mw—190,000–310,000 Da.) solution in 5% acetic acid was used as the dispersed phase. This solution was filtered using filter paper with 11 μm porosity to incorporate the active component, lignin phenol. Specifically, 150 μg of phenols were dissolved in 100 µL of Milli-Q water and added to 10.0 mL of the 1% chitosan solution. Homogeneity was promoted using vortex agitation. For the continuous phase, 150 mL of sunflower oil was taken in a 500 mL flask, and 0.1% (*w*/*w*) of Span 80 was added. Then, with a syringe, 5 mL of the dispersed phase of chitosan (F) solution was added: 1.0% drop by drop (5 mL min^−1^) on soybean oil, with constant stirring at 300 rpm or peripheral stirring, and at 30 °C. Subsequently, 20% glutaraldehyde was added four times (2.5 mL each) at 15, 30, 45, and 60 min, with constant stirring for 1 h. The concentrations of the active compounds were selected through bibliographic data [27]. The spheres were washed and dried at 40 °C and kept at 4 °C.

### 2.5. Preparation for PVA Films

First, 10 mL of 10% PVA (Acros Organics, Houston, TX, USA)) solution was prepared and dissolved in water, and then 1 mL of Spam 80 was added with vigorous agitation. Subsequently, 8 mL of glutaraldehyde (Droguería Paysandu, Paysandú, Uruguay) was slowly added (dropwise) with continuous agitation for 2 h at 30 °C. Phenol spheres were added in 3 mg/mL. The solutions were cast on a rectangular mold and maintained under constant agitation for 48 h at room temperature until solidification. The films were cut into the shape of the container lids to place an overlaid inside. Figure 1 shows the synthesis methodology for chitosan/PVA/phenol films.

### 2.6. Physical and Chemical Characterization of Phenolic Films

#### 2.6.1. Fourier-Transform Infrared Spectroscopy (FTIR)

The films’ FTIR spectra were obtained using a Nicolet 6700 spectrophotometer with a diamond ATR module (Thermo Fisher Scientific, Miami, FL, USA) in the number range waveform from 4000 to 500 cm^−1^, with a standard resolution of 4 cm^−1^ and a scanning speed of 32 cm^−1^/s. The results were analyzed using the OMNIC 8.1 software (OMNIC Series 8.1.10, Thermo Fisher Scientific).

#### 2.6.2. Thermogravimetric Analysis (TGA)

For the TGA analyses, 5 mg of sample was used and performed using a TGA Q500 (TA Instruments, New Castle, DE, USA) with a temperature ramp of 10 °C/min from 0 to 600 °C, which was subsequently ramped up to 20 °C/min from 600 °C to 1000 °C under nitrogen (flow rate 90 mL/min).

#### 2.6.3. Differential Scanning Calorimetry (DSC)

DSC was performed using the DSC Q200 equipment, TA Instruments (New Castle, DE, USA). The samples were analyzed using a ramp that covered a cycle from 25 °C to 400 °C, with a heating rate of 10 °C/min and a nitrogen flow of 10 mL/min. The data obtained were analyzed using the TA Universal Analysis software (Advantage Software v5.5.24).

#### 2.6.4. Scanning Electron Microscopy—SEM

The microspheres and the films were analyzed using a scanning electron microscope (JSM-6390LV, Jeol Lda., Tokyo, Japan) under a voltage acceleration of 10–15 kV, secondary electrons (SEI), and a spot size of 40. Images were taken at different magnifications to identify the morphology of the films. The samples were coated with a thin layer of Au in an ion cover EMS 550X Sputter Coater (Jeol Lda., Tokyo, Japan) at 20 mA for 3 min with a vacuum of 1 × 10^−1^ mbar.

#### 2.6.5. Mechanical Properties

The film’s mechanical properties were determined using a mechanical tensile tester (Shimadzu Autograph AGS-X Series, Kyoto, Japan) equipped with a 500 N load cell and pneumatic clamps to fix the samples. Five replicates (2 × 3 cm^2^) were assayed at 5 mm/min crosshead speed. The stress–strain curve was obtained, and the tensile modulus, yield strength, stress at break, and strain at break parameters were determined. The results were expressed as mean ± standard deviation.

### 2.7. Characterization of Antimicrobial Activity

#### 2.7.1. Germicidal and Germistatic Activity on Bacterial in Fresh Milk

The experiments were conducted using packaged fresh milk, adding PVA/phenol/chitosan films to assess their ability to delay milk decomposition. The experimental design focused on studying the germicidal and germistatic effects in milk, explicitly targeting the retardation of microbial growth and changes in pH and acidity. The materials included commercial fresh milk (Conaprole, Uruguay), with two setups: regular packaged fresh milk and packaged fresh milk with PVA/phenol/chitosan films.

(a)In vitro assay

Test tubes containing 10 mL of sterile milk were used. In commercial preparation, these were inoculated with 1 mL of lactic acid bacteria (Lactobacillus) and propagated at a concentration of 10 to 6. A tube was used under the same conditions, adding 1 mL of acetic acid 0.1 M as a germicide and another tube with H_2_O_2_ (0.1% H_2_O_2_, 10 vol) as a germistatic.

(b)Food trial

Commercial containers containing 1 L of milk were used. The samples were conditioned in two trials: samples in commercial containers without aggregates (X3) are white, and samples in commercial containers have a PVA/chitosan/phenol film cover (X3).

The samples were kept in identical conditions (at room temperature with a daily variation between 10 and 20 degrees Celsius), and samples were collected at regular intervals (0, 24, 48, 72, 96, and 144 h). At each point, microbial counts were performed using culture plates. After incubation, the plates were incubated at a suitable temperature and the colony-forming units (CFUs) were counted. The pH of each sample at each time point was measured using a calibrated pH meter. Acidity measurements were made in each sample using a standard titration method with a known concentration of NaOH (0.1 M) and phenolphthalein as an indicator. The CFU counts, pH values, and acidity levels at each time point for all samples were recorded, and we compared the microbial growth, pH changes, and acidity changes between the control germicidal and germistatic samples.

#### 2.7.2. Germicidal and Germistatic Activity on Yeasts in Fresh Milk

This study aims to evaluate the germicidal and germistatic effects of Saccharomyces yeast on microbial populations both in vitro and in food samples.

#### 2.7.3. In Vitro Assay

Saccharomyces cerevisiae strains were obtained from a recognized microbial culture collection at UdelaR University in Uruguay and maintained on YPD (Yeast Extract Peptone Dextrose) agar medium. To prepare yeast suspensions, the Saccharomyces yeast was grown in YPD broth at 30 °C for 24 h, and the suspension was adjusted to an optical density at 600 nm (OD_600_) of 0.1. Samples were plated on YPD agar at each designated time point to assess the germicidal effect, and colony-forming units (CFUs) were counted to determine the number of viable cells. To evaluate the germistatic effect, the OD_600_ of the samples was measured at each time point to assess growth inhibition.

Fresh milk samples were used as the dairy product in the food assay. These samples were sterilized to eliminate any pre-existing microorganisms and then inoculated with a known concentration of Saccharomyces yeast. The samples were divided into control and treated groups; the treated group received antimicrobial agents such as hydrogen peroxide (H_2_O_2_) and acetic acid. Additionally, a blank sample of fresh milk was used in its commercial container, and another sample was tested in a commercial container with an internal layer of PVA/chitosan/phenol film applied.

All inoculated samples were incubated at ambient temperatures ranging from 10 to 22 °C, and samples were collected at 0, 4, and 7 days for analysis. Microbial analysis involved performing serial dilutions of the collected samples and plating them on YPD agar media to count CFUs, which quantified the microbial load over time.

## 3. Results and Discussion

### 3.1. Preparation of PVA/Chitosan Films with Incorporation of Phenolic Microspheres

Polyvinyl alcohol (PVA), chitosan, and phenolic compounds have attracted attention for their combined potential to form films with germicidal and germistatic activities. Due to their antimicrobial properties, these films are treasured for applications in food packaging, wound dressings, and surface coatings.

Chitosan is known for its broad-spectrum antimicrobial properties, which are effective against Gram-positive and Gram-negative bacteria that can limit microbial growth by disrupting microbial cell walls and proteins. Due to its positive charge, chitosan disrupts bacterial cell membranes, leading to cell death. It also binds to bacterial DNA, inhibiting replication. When combined with phenolic compounds, which have antioxidant and antimicrobial properties, and PVA, which acts as a flexible and stable film-forming polymer, the resulting composite material shows significant germicidal and germistatic effects. This constructive collaboration between the components enhances the overall antimicrobial efficiency of the film [28,29].

One of the significant advantages of PVA/chitosan/phenolic films is their biodegradability. PVA is water-soluble and can be broken down by microorganisms under certain conditions. Chitosan is also biodegradable and originates from natural sources (chitin). The use of phenolic compounds, many of which are derived from plants, contributes to the overall eco-friendly nature of the material. This makes such films highly suitable for environmentally conscious applications, such as biodegradable packaging or coatings. One of the significant advantages of PVA/chitosan/phenolic films is their biodegradability. PVA is water-soluble and can be broken down by microorganisms under certain conditions. Chitosan is also biodegradable and originates from natural sources (chitin). The use of phenolic compounds, many of which are derived from plants, contributes to the overall eco-friendly nature of the material. This makes such films highly suitable for environmentally conscious applications, such as biodegradable packaging or coatings.

PVA is known for its excellent film-forming ability, which results in flexible, transparent, and robust films. When combined with chitosan and phenolic compounds, the mechanical properties of these films are further enhanced. Chitosan provides additional strength and flexibility due to its polymeric nature, while phenolic compounds can offer UV protection, further extending the material’s durability. Combining these materials results in a strong film with antimicrobial properties, making it multifunctional.

### 3.2. Physicochemical Characterization of Phenolic Films

#### 3.2.1. Fourier-Transform Infrared Spectroscopy (FTIR)

FTIR-ATR analysis was conducted to study the molecular interaction between PVA, chitosan, and microspheres of lignin phenols in the films. According to the results obtained (Figure 2), it was possible to observe that there are no significant differences between the spectra due to the low concentration of chitosan and phenol in the PVA matrix. Nevertheless, peaks were identified in the regions between 3250 and 3450 cm^−1^, characterized by stretching the OH group bonds in all spectra. Suflet et al. (2024) highlight that the broadband at 3437–3454 cm^−1^ is attributed to the stretching of O-H and intra- and intermolecular hydrogen bonds [30]. This is consistent with the observations made by Abdolrahimi et al. (2018) [31], who also report broad bands in the 3600–3200 cm^−1^ range associated with hydroxyl groups in PVA.

On the other hand, the peaks identified in the region between 2800 and 2950 cm^−1^ is associated with the asymmetric and symmetric overlap of CH_2_ and CH groups. This finding can be confirmed by the results of Abdou et al. (2023) and Wahab et al. (2016), which reveal that the peaks near that region can be attributed to the symmetric and asymmetric stretches of the same bonds [32,33].

In addition, the band near the range of 1725–1745 cm^−1^ related to C=O groups, as reported by Aycan et al. (2020), indicating the presence of residual functional groups that could influence the chemical reactivity of the films [34]. Furthermore, the peaks between 1627 and 1648 cm^−1^, which belong to the aromatic ring skeleton, suggest the possibility of interactions between PVA and glutaraldehyde and aromatic compounds from phenols or residues from solvents present in chitosan, as also reported by Xu et al. (2023) [35].

The bands near the ranges of 1420–1450, 1220–1240, and 1080–1150 cm^−1^ associated with the absorption peaks of CH_2_, CH, and C-O-C stretching, all characteristic of the chemical structures of chitosan and PVA. Therefore, it is expected to observe overlapping events and little differentiation [32,35]. Additionally, a peak near 850 cm^−1^ is observed, and the characteristic of C-C carbon bonds is present in polymers, as noted by Abdolrahimi et al. (2018) [31].

#### 3.2.2. Thermogravimetric Analysis (TGA)

Figure 3 shows the TGA and TGA (DTG) derivatives of PVA, PVA/Chi, PVA/Phe, and PVA/Chi/Phe films. In general, all the films show similar curves with four steps of weight loss as a function of the increasing temperature. The first step between 25 and 170 °C corresponds to the evaporation of water [35,36]. The second step was the central degradation region at 170–370 °C due to the degradation of the exposed side chain and the breakdown of polymeric chains of PVA and chitosan [37,38]. The third weight loss (370–550 °C) corresponds to the oxidative decomposition of carbon residues, and the final weight loss (above 500 °C) represents the decomposition to ash [39].

Based on the thermal profiles of the samples, there are no significant differences in the thermogram curves, except for the membrane containing chitosan, which showed a mass loss interpolation in the region of the third general event. This behavior is characteristic of the chemical interactions between PVA films and chitosan [40]. This suggests that the various additives in the formulations do not enhance the thermal stability parameters. This may be related to the additives’ low concentration and low degradation temperatures, which could influence the main decomposition events of PVA. In Table 1, it is possible to observe the intervals of events in more detail.

Table 2 shows the temperatures correlated with mass loss of 10%, 25%, and 50%. The results demonstrate significant differences in the thermal stability of the films formulated with PVA compared to those containing chitosan (Chi) and phenol (Phe).

The PVA film shows a T10 of 89.28 °C, while the PVA film with chitosan exhibits a significantly higher T10 of 116.77 °C. This suggests that chitosan contributes to more excellent thermal resistance than pure PVA. Additionally, improvements are observed in the T25 and T50 values, which reached 295.59 °C and 363.94 °C, respectively, indicating a notable enhancement in thermal stability.

On the other hand, while the PVA/Phe film also shows improvements compared to pure PVA, with T10 and T25 values of 113.73 °C and 265.77 °C, it does not reach the thermal stability levels of the chitosan films. This suggests that chitosan has a more pronounced effect on improving thermal properties than phenol.

#### 3.2.3. Differential Scanning Calorimetry

Figure 4 shows the DSC curves of the PVA, PVA/Chi, PVA/Phe, and PVA/Chi/Phe films as a function of temperature. The analysis reveals endothermic events between 75 and 175 °C, which indicate dehydration processes resulting from the evaporation of physically adsorbed water. Notably, the thermal events occurring in the range of 240 to 350 °C correspond to the melting point (Tm) and thermal pyrolysis of the PVA and chitosan polymers [40,41], aligning well with the findings from previous studies. However, in the films containing phenol, a slight shift of approximately 15 °C in the melting point was observed. This shift may be related to the chemical stability of the aromatic rings in the phenolic compounds, suggesting that their inclusion influences the thermal characteristics of the films more significantly than the other additives.

#### 3.2.4. Scanning Electron Microscopy (SEM)

In Figure 5, the micrographs of the membranes are presented in frontal view (left column) and lateral view (right column) for the samples of PVA (A and B), PVA/Chi (C and D), PVA/Phe (E and F), and PVA/Chi/Phe (G and H).

In the PVA sample (A and B), a smooth surface was observed with some small associated particles and almost no surface pores, as can also be seen in the work of Zhang et al. (2019) [42]. This contrasts with its cross-section (B), where numerous pores of different sizes and a thickness of approximately 140 µm can be seen. In contrast, the PVA/Chi film (C and D) showed an irregular surface with a scaly appearance and some cracks [43,44]. In the cross-section, measuring approximately 150 µm, a more significant number of oval pores of various sizes were observed, with some reaching up to 70 µm.

In the micrographs of the PVA/Phe sample (E and F), it was observed that the front surface (E) had a smooth appearance, with some scratches that seemed to indicate dehydration processes, as well as pores at isolated points. In general, phenols have hygroscopic properties and can interact with the polymer chains of PVA. These interactions can affect the structure of the polymer, making it more rigid. If the amount of phenols is high or not correctly distributed, the membranes can become brittle and more prone to forming cracks. Additionally, PVA/water solutions and phenol solutions have different evaporation and drying rates. If the phenol is unevenly dispersed in the film, some areas may dry faster than others, leading to differences in surface tension and crack formation in the membrane. The same applies to the other films, which can also be influenced by variations in the drying process of the different components or even by the formation of bubbles during the agitation process, providing the origin of the pores after drying. In the lateral section, a thickness of nearly 200 µm was observed, along with numerous circular pores, surpassing all other formulations. On the other hand, the PVA/Chi/Phe membrane (G and H) combined the previously mentioned characteristics, presenting a surface with alternating areas of smooth and irregular textures [45,46]. In its lateral section, the membrane had a thickness of approximately 150 µm and a moderate number of pores, some of which had quite pronounced depths.

In figures I and J, images of the chitosan microspheres with phenol are shown, displaying a heterogeneous size distribution, not exceeding 100 µm when observed individually. Additionally, it was possible to observe that the spheres have some superficial pores, possibly caused by the difference in evaporation temperatures of the solvents present. However, this appearance may be interesting as it increases their adsorption capacity and contact surface.

#### 3.2.5. Mechanical Properties

Mechanical properties are essential in food packaging to preserve product integrity, ensure durability, and maintain functionality throughout the product life cycle [47]. Blending PVA with chitosan enhances the strength and durability of films, while cross-linking with glutaraldehyde further improves their mechanical properties [48]. Additionally, incorporating particles such as phenols reinforces the films and introduces bioactivity, enhancing their overall functionality [49].

Figure 6 presents the stress–strain curve for the produced PVA/Chi/Phe film, from which key mechanical properties, including tensile modulus, yield strength, and stress and strain at break were determined. The tensile modulus indicates the film’s stiffness, yield strength marks the onset of permanent deformation, stress at break represents the maximum stress before failure, and strain at break suggests the extent of deformation at rupture.

The films exhibited a thickness of 85.80 ± 11.12 µm, reflecting good uniformity across replicates. Mechanical properties were characterized by a tensile modulus of 36.14 ± 3.73 MPa, yield strength of 12.01 ± 1.14 MPa, stress at break of 21.13 ± 0.85 MPa, and strain at break of 65.19 ± 5.96%. A similar stress–strain curve was reported in PVA/Chi films with a 65:35 ratio, where increased chitosan content led to reduced strain at break and a more rigid structure [50]. Furthermore, a recent study noted significant improvements in PVA/Chi/Phe acid films when cross-linked with glutaraldehyde, with strain at break rising from 23% to 45%, though still below the 65.19% achieved in this study. These improvements were linked to cross-linking, strengthening intermolecular bonds, and forming a robust three-dimensional network [51].

### 3.3. Characterization of Antimicrobial Activity

The control sample will exhibit average microbial growth, pH decline, and acidity increase over time. The germicidal sample should show a significant reduction in microbial growth, slower pH decline, and slower acidity increase compared to the control. The germistatic sample should show retardation in microbial growth, pH changes, and acidity changes, but to a lesser extent than the germicidal sample. These changes are shown in Table 3 (pH), Table 4 (acidity), Table 5 (Lactobacillus growth), and Table 6 (Saccharomyces growth).

For all the tests, the same milk was used. The tables show values of the zero times, to, with a deviation from the initial value of the milk. This is due to the variation produced by the addition of other germicidal and germistatic products. For the evaluation of all the tests, the real value of each one was used.

*Lactobacillus* grows well in fresh milk at optimal temperatures between 37 °C and 42 °C. Depending on the strain and environmental conditions, the exponential growth phase usually begins around 4 to 6 h after inoculation. In the test that was maintained at ambient temperature, which is lower than the optimal temperature, growth was also recorded, deteriorating the food.

Regarding pH, Table 3 shows that fresh milk has a pH close to 6.6–6.8. As *Lactobacillus* ferments lactose in milk, it produces lactic acid, which decreases the pH to 4.5–4.3. During fermentation, the pH can drop to shallow values depending on the duration and conditions of the process.

In the samples tested with acetic acid and H_2_O_2_ as positive preservation controls and the sample tested with the PVA/Chi/Phe film, it is observed that the latter controls the same as the agents known as H_2_O_2_ and controls the decrease in pH more than commercial milk, which extends the shelf life of the food. In fact, from the analysis of results, it can be seen that the food preserved with H_2_O_2_ and with the PVA/Chi/phenol film are the ones that show the least decrease in pH, coinciding with less microbial growth and maintaining the viability of the food for a longer time.

Regarding acidity in Table 4, the acidity of fresh milk is low, with a titratable acidity value of around 0.14–0.17% lactic acid. As *Lactobacillus* ferments the milk, the acidity increases, potentially reaching values of 0.8–1.0% or more lactic acid, depending on the fermentation time. These values may vary depending on the specific strain of *Lactobacillus* and culture conditions.

During storage, the pH of milk tends to decrease due to the production of lactic acid by lactic acid bacteria. This causes an increase in acidity [52,53,54]. Lactose is fermented into lactic acid, reducing its milk concentration [55,56]. Proteins can be denatured and precipitated due to increasing acidity and the action of bacterial enzymes [54]. Milk can oxidize fat, producing rancid compounds that affect flavor [57,58]. In the samples tested with acetic acid, H_2_O_2_ has a positive preservation control. In the sample tested with the PVA/Chi/Phe film, it is observed that the latter controls much more than the known agents, such as H_2_O_2,_ and controls the increase in acidity, which preserves prolonged use and only precipitates after 96 h. Summarizing, the increase in acidity occurs in all cases, due to microbial growth. In the case of the germistatic activity of H_2_O_2_ and PVA/Chi/phenol, a more controlled acidity is maintained up to 96 h, showing a very large difference from food in traditional commercial packaging.

Table 5 shows that the control sample will exhibit average microbial growth, pH decline, and acidity increase over time. The germicidal sample should show a significant reduction in microbial growth, slower pH decline, and slower acidity increase compared to the control. The germistatic sample should show retardation in microbial growth, pH changes, and acidity changes, but to a lesser extent than the germicidal sample.

The behavior of the milk kept in the original container where a PVA/chitosan/phenol film cover placed a slowdown in the physicochemical and microbiological changes and a decrease in growth, showing germistatic and germicidal activity.

In the samples evaluated with acetic acid, H_2_O_2_ has positive preservation controls, and in the sample tested with PVA/Chi/Phe film, the latter controls much more than the agents known as H_2_O_2_ and acetic acid. The decrease in the growth of Lactobacillus and the changes that this entails make it possible to prolong the shelf life by using PVA/Chi/Phe films inside the packaging since the growth in this container is 96 h at room temperature.

Yeasts are microorganisms commonly found in the environment that can contaminate and grow in fresh milk. They can enter milk during milking, processing, or storage, especially if hygienic conditions are inadequate.

These yeasts can ferment sugars in milk, such as lactose, changing the milk’s taste, smell, and texture. Additionally, yeast growth in milk can cause spoilage, making it unfit for consumption. Therefore, storing milk in refrigerated conditions and clean containers is essential to minimize the risk of contamination by yeast and other microorganisms. Table 6 shows the growth of yeast, which behaves like different microorganisms. The test was carried out for seven days, the corresponding time to observe yeast growth.

CFU counts and OD600 values should be recorded at each time point. The growth and survival rates between the control and treated groups should be compared. Any changes in the sensory and physicochemical properties of the food samples, such as precipitate and phase separation, should be analyzed.

In the tests, yeasts contaminated 70% of commercially packaged milk. When acetic acid and H_2_O_2_ are used in the formulation, there is control, but the organoleptic qualities quickly change, and separation appears or is impossible to measure. On the other hand, when the film is used in part of the packaging and is not in contact with the food, there is significant control with low yeast growth and preservation for a week.

In the dairy industry, phenols can control the growth of pathogenic and spoilage bacteria in products such as milk and cheese. However, its use must be carefully regulated due to its potential to alter the flavor and sensory quality of the final product.

Some microorganisms may develop resistance to phenols, which could limit their long-term effectiveness. Additionally, it is essential to consider the impact of phenols on the beneficial microbiota in fermented products such as yogurt and kefir.

Phenols can impart undesirable flavors and odors to dairy products, which could affect their acceptability to consumers. This is especially critical in products with flavor, such as fine cheeses, a crucial attribute.

In this work, we worked with natural lignin polyphenols. The proposal has been to use them in packaging without introducing them into the food formula. Therefore, the effects on flavor and texture, among others, are not altered. This is an essential aspect since food formulations are not modified, but in this case, a control is performed from the lid of the container.

The degradation behavior of the developed PVA/chitosan/phenol films is also a critical consideration for their application in sustainable packaging. The literature indicates that pure PVA films exhibit limited biodegradation, with approximately 3.94% degradation observed over 14 days in soil [59]. The degradation rate is influenced by environmental conditions and the presence of specific microorganisms capable of breaking down PVA. Chitosan-based films degrade within 4 weeks under natural composting conditions, using soil microorganisms [60]. Adding chitosan to PVA films significantly improves their biodegradability. PVA/chitosan composite films demonstrate enhanced properties compared to pure PVA films, including improved mechanical strength, chemical resistance, and biodegradability [58,59,60,61].

While phenolic compounds may exhibit antimicrobial effects, PVA/chitosan films remain biodegradable. The addition of natural extracts to these films can influence their properties, including water resistance and mechanical properties. However, the incorporation of these compounds does not significantly impair the films’ overall biodegradability. In fact, chitosan-based films with added natural extracts have shown excellent biodegradability in soil environments, with some formulations completely degrading within 6–8 weeks [60].

These films offer several advantages, such as enhanced antimicrobial activity, the biodegradability of all their components, controlled release properties, and non-toxicity. These characteristics make them particularly useful in medical, packaging, and environmental applications, including the degradability of the films and their potential effects on other microbial communities. These are areas of interest for the research group for future investigations.

## 4. Conclusions

From the tests of synthesis of phenolic units, the production of phenol/chitosan microspheres, and their incorporation into PVA film formulations, we can say that new films with good physicochemical, mechanical, and germicidal properties were obtained, which allows us to conclude that we are dealing with a product applicable to the preservation of food in its packaging. From the analysis of results, it can be seen that the food preserved with H_2_O_2_ and with the PVA/Q/phenol film are the ones that show the least decrease in pH, coinciding with less microbial growth and maintaining the viability of the food for a longer time. We can see that the increase in acidity occurs in all cases, due to microbial growth. In the case of the germ static activity of H_2_O_2_ and PVA/Q/phenol, a more controlled acidity is maintained up to 96 h, showing a very large difference with food in traditional commercial packaging.

It is known that phenols have considerable potential as germicidal agents in dairy products, but their use must be handled with caution. It is crucial to balance their antimicrobial capacity with the possible adverse effects on the quality of the product, including changes in flavor. In this case, to avoid these problems, the application of the controlling agents chitosan and phenols were not incorporated into the food but in the lid of the container (in the form of membranes on the back lid) creating an environment to control the natural microbial growth of the food (in this case fresh milk), as well as the control of the pH and acidity that appear as deterioration of the food due to microbial growth and no different aromas were observed, probably because they were encapsulated.

For all these reasons, we can conclude that this material can result in a solution for the control of food deterioration, with its biological composition being fundamental, safe with food, and that in this case it is not within the formulation of the food itself, as it is with acetic acid and hydrogen peroxide tested in this work.

In future works, details of the residence time of the germicidal action will be adjusted in normal and adverse situations in the form of food preservation, and the tests will be extended to other foods, both liquid and solid.

## Figures and Tables

**Figure 1 polymers-17-00082-f001:**
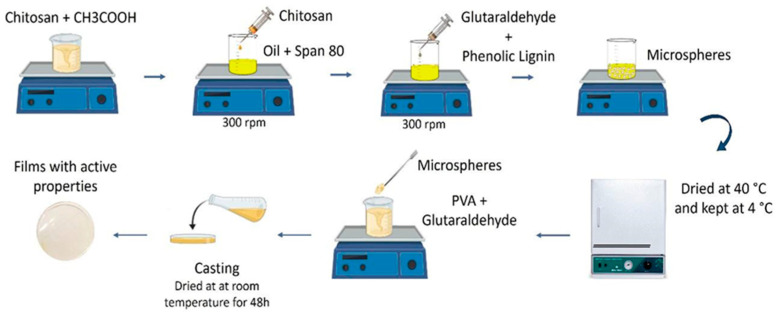
Outline of the synthesis methodology for chitosan/PVA/phenol films.

**Figure 2 polymers-17-00082-f002:**
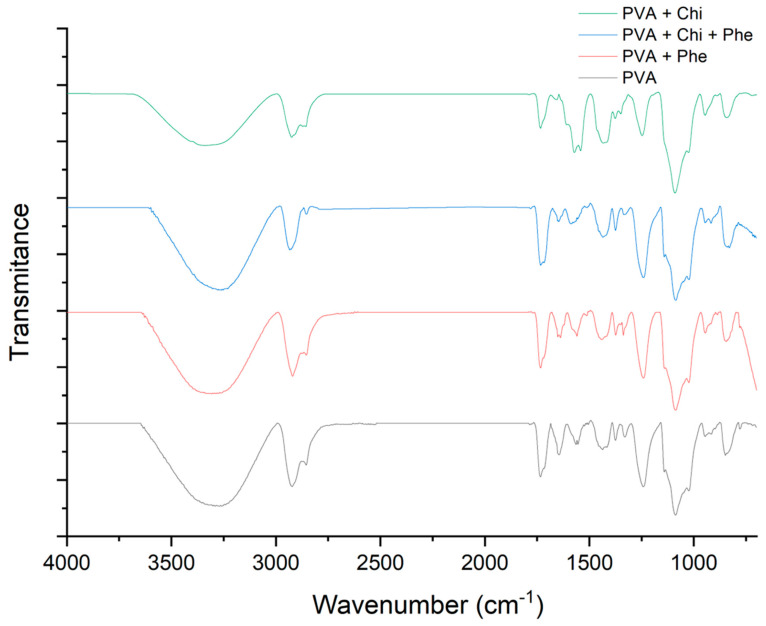
FTIR spectrum of PVA/Chi, PVA/Chi/Phe, PVA/Phe, and pure PVA samples.

**Figure 3 polymers-17-00082-f003:**
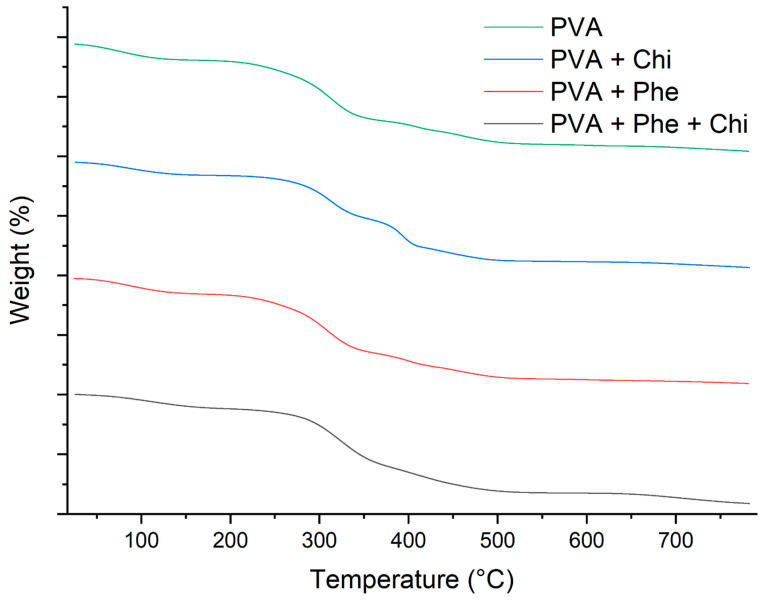
TGA curves of the samples of films formed by PVA and combinations under a nitrogen atmosphere.

**Figure 4 polymers-17-00082-f004:**
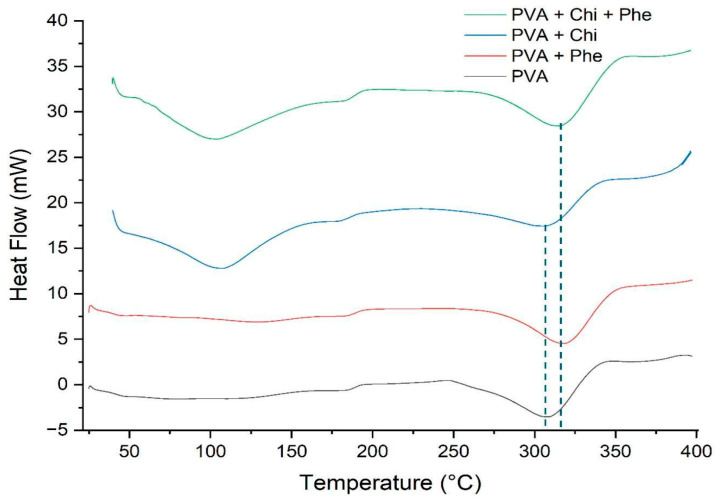
DSC curves of the samples of films formed by pure PVA and PVA with additives. The dash lines put evidence of the shift in Tm (melting point) for the sample PVA + Chi + Phe.

**Figure 5 polymers-17-00082-f005:**
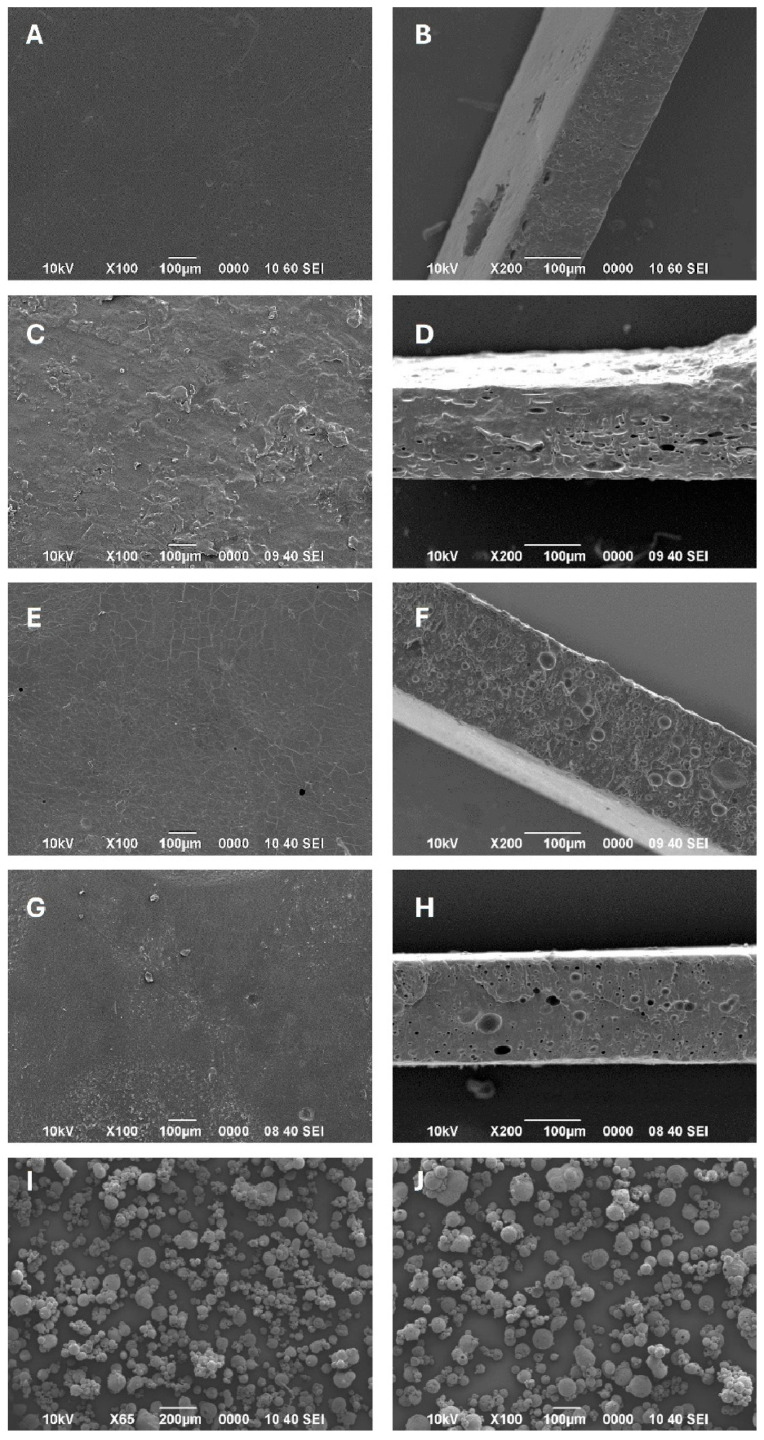
SEM electromicrographs of the samples of PVA (**A**,**B**), PVA/Chi (**C**,**D**), PVA/Phe (**E**,**F**), PVA/Chi/Phe (**G**,**H**), and phenol microspheres (**I**,**J**).

**Figure 6 polymers-17-00082-f006:**
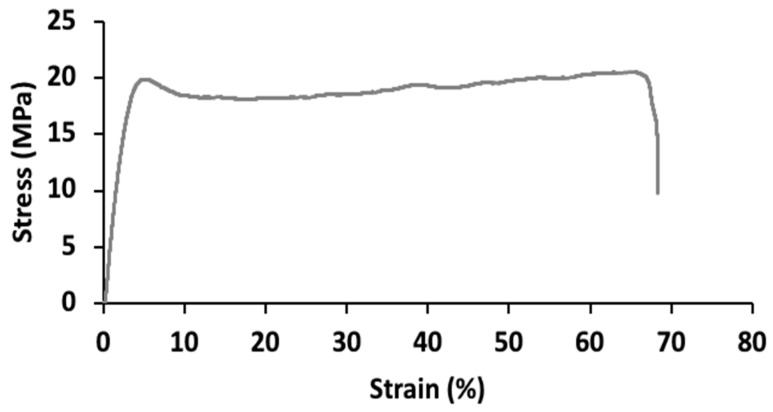
Stress–strain curve obtained for the PVA/Chi/Phe composite film.

**Table 1 polymers-17-00082-t001:** Temperature intervals of the main mass loss events were observed for the evaluated films.

SAMPLES	1° Event	2° Event	3° Event	4° Event
PVA	25–161	161–373	373–549	549–650
PVA/Chi	25–177	177–355	355–420	420–600
PVA/Phe	25–174	174–366	366–569	569–644
PVA/Chi/Phe	25–196	196–383	383–581	581–750

**Table 2 polymers-17-00082-t002:** Table of temperatures associated with 10%, 25%, and 50% mass loss in PVA films and PVA with additives.

SAMPLES	T10 (°C)	T25 (°C)	T50 (°C)	Residue at 750 °C (%)
PVA	89.28	259.73	318.5	11.13
PVA/Chi	116.77	295.59	363.94	12.63
PVA/Phe	113.73	265.77	319.35	12.53
PVA/Chi/Phe	139.5	299.52	340.9	10.12

**Table 3 polymers-17-00082-t003:** Evaluation of pH changes observed in in vitro food tests.

Sample	0 h	24 h	48 h	72 h	96 h	144 h	192 h
Sterile inoculated sample	6.47 ± 0.06	5.07 ± 0.12	4.47 ± 0.06	4.13 ± 0.12	curdled	curdled	-
Sample with acetic acid	6	5.3 ± 0.10	4.9 ± 0.10	3.97 ± 0.06	curdled	curdled	-
Sample with H_2_O_2_	6.4	5.77 ± 0.06	5.47 ± 0.06	4.77 ± 0.06	4.2	curdled	-
Commercial packaging	6.23 ± 0.06	5.13 ± 0.12	4.8 ± 0.20	4.3 ± 0.2	curdled	curdled	-
Packaging with PVA/Chi/Phe	6.27 ± 0.06	5.77 ± 0.06	5	4.93 ± 0.06	4.77 ± 0.06	4.5 (thick)	curdled

**Table 4 polymers-17-00082-t004:** Evaluation of acidity change in in vitro food tests.

Sample	0 h	24 h	48 h	72 h	96 h	144 h
Sterile inoculated sample	0.18	0.41 ± 0.01	0.76 ± 0.01	0.95 ± 0.03	Heterogeneous sample	Precipitated
Sample with acetic acid	0.15	0.35 ± 0.01	0.50 ± 0.02	0.64 ± 0.23	Heterogeneous sample	Precipitated
Sample with H_2_O_2_	0.16	0.22 ± 0.02	0.29 ± 0.01	0.39 ± 0.01	0.53 ± 0.03	Precipitated
Commercial packaging	0.15	0.2	0.25 ± 0.01	0.80 ± 0.01	Heterogeneous sample	Precipitated
Packaging with PVA/Chi/Phe	0.15	0.25	0.3	0.45 ± 0.03	0.53 ± 0.03	Precipitated

**Table 5 polymers-17-00082-t005:** In vitro and food assays—Microbial growth Lactobacillus.

Sample	0 h	24 h	48 h	72 h	96 h	144 h
Sterile inoculated sample	Five colonies	24.67 ± 2.52 colonies	42.67 ± 2.31 colonies	30% plate	Heterogeneous sample	Precipitated
Sample with acetic acid	0	25.6 ± 1.15 colonies	-	-	Heterogeneous sample	Precipitated
Sample with H_2_O_2_	0	7.67 ± 2.52 colonies	20.67 ± 1.15 colonies	25% plate	85% plate	Precipitated
Commercial packaging	4.33 ± 0.58 colonies	10.67 ± 1.15 colonies	50% plate	65% plate	Heterogeneous sample	Precipitated
Packaging with PVA/Chi/Phe	4.33 ± 0.58 colonies	5.33 ± 2.52 colonies	40.33 ± 4.51 colonies	28% plate	80% plate	Precipitated

**Table 6 polymers-17-00082-t006:** In vitro and food assays. Changes in microbial growth in 7 days of culture and CFU counts and OD600 nm values.

Sample	CFU t0	OD 600 nm t0	CFU t4	OD 600 nm t4	CFU t7	OD 600 nm t8
Sterile inoculated sample	5	0.064 ± 0.005	60% Plate	Precipitate	-	-
Sample with acetic acid	3	0.074 ± 0.003	45% Plate	-	-	-
Sample with H_2_O_2_	4	0.046 ± 0.002	-	Separation	-	-
Commercial packaging	2	0.094 ± 0.004	70% Plate	Separation	-	-
Packaging with PVA/Chi/Phe	3	0.031 ± 0.003	30% Plate	40% Plate	0.11	Separation

## Data Availability

The original contributions presented in the study are included in the article, further inquiries can be directed to the corresponding author.

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
