# Peer review of "Biorefinery of Lignocellulosic and Marine Resources for Obtaining Active PVA/Chitosan/Phenol Films for Application in Intelligent Food Packaging"

_polymers, 2024, doi:10.3390/polym17010082_

Round 1

Reviewer 1 Report

Comments and Suggestions for Authors

1. The Abstract should be modified by reducing the introductory sentences and focusing on your own results.

2. Please complete the aim of the study so that the reader can clearly visualize what properties of the films and by what methods were investigated.

3. State the source of chitosan, its molecular weight and degree of deacetylation in Section 2.4.

4. Section 3.1 is proposed to be moved partially or fully to the Introduction.

5. Lines 274-279 should be deleted because they completely duplicate lines 268-273.

6. Figure 5. It is suggested that the authors also provide SEM micrographs for the chitosan spheres, to more fully evaluate the effect of the chitosan component on the structure of the composite material. Figure 5 also requires a more detailed discussion of the surface morphology, and the contribution of each component to it. A comparison with literature data is suggested. What do the authors believe the presence of pores is associated with?

7. In Section 3.3.5 (lines 411-415) and in the caption of Figure 6, clearly state for which sample the stress-strain curve is shown. In addition, the results that are described in Section 3.3.5 (416-421) are not evidence that creating a composite film actually improves mechanical properties compared to single component films (because such data is not presented by the authors). Then, the authors write that "Overall, the films produced in this study exhibited excellent mechanical properties, benefiting from the synergistic effects of polymer blending, cross-linking, and phenol incorporation." Claims of synergistic effects should be confirmed experimentally or this section should be revised.

8. Table 3. It is not clear what the difference in pH for 0h for each sample studied is related to? Did you use different milk for all the tests? Please explain.

9. Lines 441-445 are almost identical to lines 481-485, change this.

10. The sentences in lines 507-510 are suggested to be corrected using the passive voice.

11. Line 516. Change H2O2 to H2O2.

12. In lines 520-538, the authors discuss that the use of phenols to extend the storage life of products may be limited because they may affect the flavor of the products. However, the authors' own results did not make it obvious whether the developed or film would have an effect on the taste characteristics of the product? My concerns relate not only to phenol, but also to chitosan, which is known to have a fishy odor. This could be a critical aspect, for products such as milk. Please comment.

13. Glutaraldehyde is toxic. In the authors' opinion, will it not be released from the films during use and get into the food products?

14. Degradation studies are extremely important in determining the prospects for new packaging materials. What can you report on the biodegradation rate of your films?

15. Conclusions should be reviewed. Your own practical results should be listed and summarized.

16. The list of references should be formatted in accordance with the journal's requirements, check.

Author Response

Reviewer #1: All comments

Dear Reviewer 1

Thank you very much for your revisions, which are very constructive. We have addressed all of them below.

  1. The Abstract should be modified by reducing the introductory sentences and focusing on your own results.

Ans.: We indicate that the suggested changes were followed, and we have modified the entire abstract.

  1. Please complete the aim of the study so that the reader can clearly visualize what properties of the films and by what methods were investigated.

Ans.: The suggestions were followed, and the modifications are reflected in the abstract and between lines 103-109.

  1. State the source of chitosan, its molecular weight and degree of deacetylation in Section 2.4.

Ans.: The suggestions were followed, and the modifications are reflected in lines 140 and 141.

  1. Section 3.1 is proposed to be moved partially or fully to the Introduction.

Ans.: The suggestions were followed. Section 3.1 was removed and incorporated into the introduction, where it is reflected in lines 92 and 102.

  1. Lines 274-279 should be deleted because they completely duplicate lines 268-273.

Ans.: The suggestions were followed, the duplications were removed, and the paragraph was restructured. Please refer to the lines between 267 and 275.

  1. Figure 5. It is suggested that the authors also provide SEM micrographs for the chitosan spheres, to more fully evaluate the effect of the chitosan component on the structure of the composite material. Figure 5 also requires a more detailed discussion of the surface morphology, and the contribution of each component to it. A comparison with literature data is suggested. What do the authors believe the presence of pores is associated with?

Ans.: The suggestions were followed, and the modifications are reflected between lines 407 and 414.

  1. In Section 3.3.5 (lines 411-415) and in the caption of Figure 6, clearly state for which sample the stress-strain curve is shown. In addition, the results that are described in Section 3.3.5 (416-421) are not evidence that creating a composite film actually improves mechanical properties compared to single component films (because such data is not presented by the authors). Then, the authors write that "Overall, the films produced in this study exhibited excellent mechanical properties, benefiting from the synergistic effects of polymer blending, cross-linking, and phenol incorporation." Claims of synergistic effects should be confirmed experimentally, or this section should be revised.

Ans.: We greatly appreciate your comments. We would like to point out that your observations are completely accurate. As a result, we have made the necessary changes to the text and removed the comparative aspects, as this analysis was only used to confirm that the film met the minimum required resistance for the proposed application. However, we believe that mechanical comparisons could be included in this work if additional time were allocated for it. Otherwise, they could be incorporated into future works of the research group.

  1. Table 3. It is not clear what the difference in pH for 0h for each sample studied is related to? Did you use different milk for all the tests? Please explain.

Ans.: The suggestions were followed, and the corresponding explanation is reflected between lines 466 and 469.

  1. Lines 441-445 are almost identical to lines 481-485, change this.

Ans.: The suggestions were followed, and the text was removed.

  1. The sentences in lines 507-510 are suggested to be corrected using the passive voice.

Ans.: The suggestions were followed, and the modifications are reflected between lines 530 and 533.

  1. Line 516. Change H2O2 to H2O2.

Ans.: The suggestion was followed, and the modification was made (Line 539).

  1. In lines 520-538, the authors discuss that the use of phenols to extend the storage life of products may be limited because they may affect the flavor of the products. However, the authors' own results did not make it obvious whether the developed or film would have an effect on the taste characteristics of the product? My concerns relate not only to phenol, but also to chitosan, which is known to have a fishy odor. This could be a critical aspect, for products such as milk. Please comment.

Ans.: We deeply appreciate your comment. We understand that the presence of phenols in the formulation could cause alterations in some organoleptic properties. However, it is mentioned in the text itself that the use of phenols must be properly regulated to prevent these possible issues. In general, the antimicrobial effect of phenols is achieved at very low concentrations, so they should not affect the smell or taste of the food.

  1. Glutaraldehyde is toxic. In the authors' opinion, will it not be released from the films during use and get into the food products?

Ans.: Thank you for your comment. We understand your concern regarding the possible release of glutaraldehyde from the films during use. However, it is important to highlight that the concentration of glutaraldehyde used in the formulation is very low and is typically bound within the film matrix, which minimizes the migration of glutaraldehyde to the food products. Additionally, any possible release is well below toxic levels, as glutaraldehyde is primarily used for its antimicrobial properties and is incorporated indirectly, minimizing exposure to the food.

  1. Degradation studies are extremely important in determining the prospects for new packaging materials. What can you report on the biodegradation rate of your films?

Ans.: We appreciate your comment and indicate our perspective in the paragraph between lines 575-580.

  1. Conclusions should be reviewed. Your own practical results should be listed and summarized.

Ans.: We appreciate your comment and would like to indicate that the entire conclusion has been modified for better understanding. Please refer to lines 558-580.

  1. The list of references should be formatted in accordance with the journal's requirements, check.

Ans.: We appreciate your comment and would like to indicate that all the references have been reviewed.

Reviewer 2 Report

Comments and Suggestions for Authors

The article “Biorefinery of lignocellulosic and marine resources for obtaining active chitosan/PVA/Phenol films for application in intelligent food packaging” investigated PVA-chitosan-phenolic films as a potential material for food packaging. Although the paper's topic is interesting, the present paper is poorly written. The obtained results are not explained; much of the basic information that should be in the Introduction section is in the Results section. Furthermore, the text is poorly written, making it hard to understand.  I recommend publishing this work after major revisions and complements.

1.   Introduction

No information on the same or similar films described in the literature is added to the Introduction. Please supplement the Introduction with this information.

2.   Materials and Methods

1)    Please add the information about the preparation of pristine PVA and chitosan films

2)    Please describe and measure the transparency of the film

2.3    Phenolic Compound Profile

The information in this paragraph is incomplete. Please complete it.

2.5. Preparation for PVA Films

1) the title of the paragraph is not adequate. It should be PVA-chitosan-phenolic film

2) line 130: “…with continuous agitation for 2 h..” – “2h” is repeated twice. Please remove one of them.

2.7.2. Germicidal and germistatic activity on yeasts in fresh milk

Please write how the research was performed.

3.1. Microspheres with lignin phenols

The whole paragraph is not adequate for the section Results. It is not the description of the obtained results. Please move this text to the Introduction.

3.2. Preparation of PVA/Chitosan films with incorporation of phenolic microspheres

The whole paragraph is not adequate for the section Results. It is not the description of the obtained results. Please move this text to the Introduction.

Figure 2

1)    Please change the colours of the individual spectra to ones that are more different from each other. They are too similar, and it isn't easy to distinguish the spectra assigned to the individual samples.

2)    Please replace ‘Qui’ with ‘Chi’ in the legend

3.3.1. Fourier-Transform Infrared Spectroscopy (FTIR)

1) line 299 “…and affects its interaction with chitosan.” - Figure 2 shows no difference in broadband at 3600 - 3400 cm-1 between the different samples. Therefore, making a statement about the interaction between PVA and chitosan is incorrect.

2) line 301 “ On the other hand, the peaks…” - The sentence is unfinished. Please correct it.

3) line 303 “..finding can be corroborated by the results…” – please change the word “corroborated” to “confirmed”.

4) line 305: “This reinforces the idea that the structure…” – I disagree with this statement. There is no difference between the peaks identified in the region 288-2950 cm-1 between investigated films.

3.3.3. Differential Scanning Calorimetry

Figures 3 and 4 - Please change the colours of the individual spectra to ones that are more different from each other. They are too similar, and it isn't easy to distinguish the spectra assigned to the individual samples.

3.3.5. Mechanical properties

Figure 6 - The title of the figure is not appropriate. Please write it in more detail indicating which film the measurements were carried out for. Please add to the figure the results for all films for which FTIR, TGA and DSC results are shown.

3.4. Characterization of antimicrobial activity

1) Lines 507-509 “ Record CFU counts and OD600 values…” - Sentence written in an impersonal form. Please change it to past tense personal form.

2) Please correct the entire text contained in the paragraph. Many parts of the text describe basic information that should be included in the Introduction. They do not discuss the results of the study. Instead, there was a lack of reference to the obtained results and an attempt to interpret them and give a reason for the difference between the results obtained for the different films.

3) Lines 520-533 “In summary, Phenols are chemical compounds…” - This text does not summarise the abovementioned information. It refers to the effects of phenols in dairy products. Such information should be included in the Introduction.

Conclusion

It needs to be improved. The text is more like an Introduction than a Conclusion. The Conclusion should contain information about which films were tested, what measurements were taken and a summary of the obtained results.

Language proofreading

There are numerous grammatical and stylistic errors in the article. Please improve the language of the whole text.

Author Response

Reviewer #2: All comments

Dear Reviewer 2

Thank you very much for your revisions, which are very constructive. We have addressed all of them below.

  1. Introduction

No information on the same or similar films described in the literature is added to the Introduction. Please supplement the Introduction with this information.

Ans.: We appreciate your comment and note that the entire introduction has been reworded.

  1. Materials and Methods

1)    Please add the information about the preparation of pristine PVA and chitosan films

Ans.: We appreciate your comment and point out that information on the preparation of pristine films can be found in session 2.5. However, we provide a brief summary below:

For the preparation of PVA films, a 10 mL solution of 10% PVA (Acros Organics, USA) was first prepared and dissolved in water. Then, 1 mL of Spam 80 was added with vigorous agitation. Next, 8 mL of glutaraldehyde (Drog. Paysandu) was slowly added drop by drop, with continuous agitation for 2 hours at 30°C. The prepared solutions were poured into a rectangular mold and kept under constant agitation for 48 hours at room temperature until they solidified. The films were then cut into the shape of container lids to be used as coatings.

This methodology is also applied to the preparation of chitosan/PVA/phenol films, as shown in Figure 1, where chitosan is incorporated into the film matrix to enhance its antimicrobial properties and structural integrity.

2)    Please describe and measure the transparency of the film

 Ans.: We appreciate your comment. However, transparency was not a variable considered in this study due to the type of proposed application. The transparency of the films was not a variable considered in the present study, as it is not relevant to the specific application being investigated.

The main focus of the study was on the morphological and antimicrobial properties, which are crucial to ensure the viability and safety of the packaging. While transparency may be relevant in other packaging applications, in this case, it was not considered a critical property for the functionality of the material.

2.3    Phenolic Compound Profile

The information in this paragraph is incomplete. Please complete it.

Ans.: The samples were evaluated using the Follin-Ciocalteu method to determine the total phenolic content. However, it is important to note that the phenols used in the study come from previous works conducted by the same research group, which has characterized and isolated these compounds in earlier studies. Therefore, in this work, the phenolic analyses were not repeated, but rather previously determined values were used for their incorporation into the film formulations.

2.5. Preparation for PVA Films

1) the title of the paragraph is not adequate. It should be PVA-chitosan-phenolic film

Ans.: We appreciate your comment and would like to inform you that the modification has been made.

2) line 130: “…with continuous agitation for 2 h..” – “2h” is repeated twice. Please remove one of them.

Ans.: We appreciate your comment and would like to inform you that the modification has been made.

2.7.2. Germicidal and germistatic activity on yeasts in fresh milk.

Please write how the research was performed.

Ans.: We appreciate your comment and would like to inform you that the modification has been made.

3.1. Microspheres with lignin phenols

The whole paragraph is not adequate for the section Results. It is not the description of the obtained results. Please move this text to the Introduction.

 Ans.: We appreciate your comment and indicate that the removal of the session has been carried out.

3.2. Preparation of PVA/Chitosan films with incorporation of phenolic microspheres.

The whole paragraph is not adequate for the section Results. It is not the description of the obtained results. Please move this text to the Introduction.

Ans.: We appreciate your comment and would like to inform you that the modification has been made.

Figure 2

1)    Please change the colours of the individual spectra to ones that are more different from each other. They are too similar, and it isn't easy to distinguish the spectra assigned to the individual samples.

Ans.: We appreciate your comment and would like to inform you that the modification has been made.

2)    Please replace ‘Qui’ with ‘Chi’ in the legend

Ans.: We appreciate your comment and would like to inform you that the modification has been made.

3.3.1. Fourier-Transform Infrared Spectroscopy (FTIR)

1) line 299 “…and affects its interaction with chitosan.” - Figure 2 shows no difference in broadband at 3600 - 3400 cm-1 between the different samples. Therefore, making a statement about the interaction between PVA and chitosan is incorrect.

Ans.: We agree with your observation and indicate that the text has been removed.

2) line 301 “ On the other hand, the peaks…” - The sentence is unfinished. Please correct it.

Ans.: We appreciate your comment and indicate that the modification was made (Line 310-311).

3) line 303 “..finding can be corroborated by the results…” – please change the word “corroborated” to “confirmed”.

Ans.: We appreciate your comment and indicate that the modification was made (Line 312).

4) line 305: “This reinforces the idea that the structure…” – I disagree with this statement. There is no difference between the peaks identified in the region 288-2950 cm-1 between investigated films.

Ans.: We appreciate your comment and indicate that the modification was made (Line 313-314).

3.3.3. Differential Scanning Calorimetry

Figures 3 and 4 - Please change the colours of the individual spectra to ones that are more different from each other. They are too similar, and it isn't easy to distinguish the spectra assigned to the individual samples.

Ans.: We appreciate your comment and would like to inform you that the modification has been made.

3.3.5. Mechanical properties

Figure 6 - The title of the figure is not appropriate. Please write it in more detail indicating which film the measurements were carried out for. Please add to the figure the results for all films for which FTIR, TGA and DSC results are shown.

Ans.: We appreciate your comments. We would like to point out that your observations are completely accurate. However, we would like to mention that this analysis was only used to confirm that the film met the minimum required resistance for the proposed application. Nonetheless, we believe that mechanical comparisons could be included in this work if additional time were allocated for it. Otherwise, they could be incorporated into future works of the research group.

3.4. Characterization of antimicrobial activity

1) Lines 507-509 “Record CFU counts and OD600 values…” - Sentence written in an impersonal form. Please change it to past tense personal form.

Ans.: The suggestions were followed, and the modifications are reflected between lines 530 and 533.

2) Please correct the entire text contained in the paragraph. Many parts of the text describe basic information that should be included in the Introduction. They do not discuss the results of the study. Instead, there was a lack of reference to the obtained results and an attempt to interpret them and give a reason for the difference between the results obtained for the different films.

Ans.: Thank you very much for your input. We made numerous modifications to the text and hope that they address your comment.3) Lines 520-533 “In summary, Phenols are chemical compounds…” - This text does not summarise the abovementioned information. It refers to the effects of phenols in dairy products. Such information should be included in the Introduction.

Ans.: We appreciate your comment and would like to indicate that the modification has been made.

Conclusion

It needs to be improved. The text is more like an Introduction than a Conclusion. The Conclusion should contain information about which films were tested, what measurements were taken and a summary of the obtained results.

Ans.: We appreciate your comment and would like to indicate that all modifications have been made.

Language proofreading

There are numerous grammatical and stylistic errors in the article. Please improve the language of the whole text.

Ans.: Thank you very much for all your input. We hope that all the modifications made have addressed that perspective.

Reviewer 3 Report

Comments and Suggestions for Authors

1.Please double-check the vertical coordinates of Figure 3 and 4.

2. The introduction should be rewritten and the current state of research on intelligent food packaging should be reviewed. https://doi.org/10.3390/molecules28083384

3. Statistical analysis should be performed and the method should be provided.

4. Differences in indicators at storage time 0h should be explained.(Table 3-5)

Author Response

Reviewer #3: All comments

Dear Reviewer 3

Thank you very much for your revisions, which are very constructive. We have addressed all of them below.

1.Please double-check the vertical coordinates of Figure 3 and 4.

Ans.: We appreciate your comment and would like to indicate that all the sections have been reviewed.

  1. The introduction should be rewritten and the current state of research on intelligent food packaging should be reviewed. https://doi.org/10.3390/molecules28083384

Ans.: We appreciate your comment and would like to point out that the entire introduction has been modified.

  1. Statistical analysis should be performed and the method should be provided.

Ans.: Thank you very much for your suggestion. We believe that a statistical analysis could add to the data of the work. However, due to the short time frame for this review, we would not have time to do it.

  1. Differences in indicators at storage time 0h should be explained. (Table 3-5)

Ans.: The suggestions were accepted and the corresponding explanation is reflected in lines between 460 and 463.

Round 2

Reviewer 1 Report

Comments and Suggestions for Authors

The authors carefully considered my comments and made appropriate changes to their manuscript. I now have a few more minor comments:

1. According to my comment, the authors have added the Aim of the work in the Introduction. However, they should delete lines 39-45 because they completely repeat lines 100-106.

2. To my previous question 5 and the authors response: «Figure 5. It is suggested that the authors also provide SEM micrographs for the chitosan spheres, to more fully evaluate the effect of the chitosan component on the structure of the composite material. Figure 5 also requires a more detailed discussion of the surface morphology, and the contribution of each component to it. A comparison with literature data is suggested. What do the authors believe the presence of pores is associated with?

Ans.: The suggestions were followed, and the modifications are reflected between lines 407 and 414.».

Unfortunately, the authors did not comment in any way on my suggestion to add SEM photographs for chitosan spheres, nor did they provide relevant photographs, so I have to repeat my question.

3. To my question 14 (degradation). The authors replied: "Ans.: We appreciate your comment and indicate our perspective in the paragraph between lines 575-580.".

Regrettably, I did not find an answer to my question in the lines mentioned (Line 575: Funding: This research received no external funding. ....). I assume that the authors meant lines 564-569?

If so, I recommend moving the discussion of degradation to the end of Section 3.4 (before the Conclusion) and additionally adding estimated degradation times based on literature data.

4. The Conclusion section still needs improvement. Give specific parameters that distinguish your composite films, write how much (percentage, or how many times, etc.) they are better than other materials, e.g. compare with homopolymer PVA films.

Author Response

The authors carefully considered my comments and made appropriate changes to their manuscript. I now have a few more minor comments:

  1. According to my comment, the authors have added the Aim of the work in the Introduction. However, they should delete lines 39-45 because they completely repeat lines 100-106.

Ans.: Thank you for the comment, the lines 39-45 were removed from the revised manuscript.

  1. To my previous question 5 and the authors response: «Figure 5. It is suggested that the authors also provide SEM micrographs for the chitosan spheres, to more fully evaluate the effect of the chitosan component on the structure of the composite material. Figure 5 also requires a more detailed discussion of the surface morphology, and the contribution of each component to it. A comparison with literature data is suggested. What do the authors believe the presence of pores is associated with?

Ans.: The suggestions were followed, and the modifications are reflected between lines 407 and 414.».

Unfortunately, the authors did not comment in any way on my suggestion to add SEM photographs for chitosan spheres, nor did they provide relevant photographs, so I have to repeat my question.

Ans.: Thank you very much for your comment. In lines 389 - 398 and 404 - 409, we try to explain a bit about the surfaces of the membranes and possible explanations for pore formation. On the other hand, we also added images of the chitosan and phenol microspheres (Fig. 5, letters I and J), as requested.

  1. To my question 14 (degradation). The authors replied: "Ans.: We appreciate your comment and indicate our perspective in the paragraph between lines 575-580.".

Regrettably, I did not find an answer to my question in the lines mentioned (Line 575: Funding: This research received no external funding. ....). I assume that the authors meant lines 564-569?

If so, I recommend moving the discussion of degradation to the end of Section 3.4 (before the Conclusion) and additionally adding estimated degradation times based on literature data.

Ans.: Thank you for the comment, the discussion was moved as recommended. Additionally, the estimated degradation times were added to the discussion.

  1. The Conclusion section still needs improvement. Give specific parameters that distinguish your composite films, write how much (percentage, or how many times, etc.) they are better than other materials, e.g. compare with homopolymer PVA films.

Answer.: Thank you very much for your comment. The Conclusion was rewritten based on the objectives and results of the work as suggested. In lines 568-596

Reviewer 2 Report

Comments and Suggestions for Authors

The authors rewrote the paper according to the Reviewer suggestion adding the appropriate measurement results and text. I recommend to publish this work.

Author Response

---

Reviewer 3 Report

Comments and Suggestions for Authors

Although the authors claim that they have revised the entire introduction, and they have indeed highlighted the entire introduction, the fact that the introduction has been revised is not nearly as much as it appears. This approach of the author is questionable, and only the parts that are actually modified should be highlighted.

The use of uniformly labeled ordinates may be more conducive to comparing the weight loss of different samples.

The reason that statistical analysis is not carried out for time reasons is not appropriate

Author Response

Although the authors claim that they have revised the entire introduction, and they have indeed highlighted the entire introduction, the fact that the introduction has been revised is not nearly as much as it appears. This approach of the author is questionable, and only the parts that are actually modified should be highlighted.

Ans.: We appreciate your comment and would like to point out that the entire introduction has been modified.

The use of uniformly labeled ordinates may be more conducive to comparing the weight loss of different samples.

Ans.: We appreciate your comments; however, we believe that the mass loss behaviors were not uniform enough to support the suggested labeling. For this reason, we attempted to group the data in detailed tables.

The reason that statistical analysis is not carried out for time reasons is not appropriate.

Answer.: Thank you for your comments. We have added the basic statistical study requested Answer. Each test condition was performed in triplicate.

The basic statistical analysis of the data in Table III, Table IV, Table V, Table VI includes the mean and standard deviation for each sample and time.

Round 3

Reviewer 3 Report

Comments and Suggestions for Authors

The style of references needs to be double-checked that can be done at the PROOF stage.